# Physicochemical and Sensorial Evaluation of Meat Analogues Produced from Dry-Fractionated Pea and Oat Proteins

**DOI:** 10.3390/foods9121754

**Published:** 2020-11-27

**Authors:** Davide De Angelis, Aleksei Kaleda, Antonella Pasqualone, Helen Vaikma, Martti Tamm, Mari-Liis Tammik, Giacomo Squeo, Carmine Summo

**Affiliations:** 1Department of Soil, Plant and Food Science (DISSPA), University of Bari Aldo Moro, Via Amendola, 165/a, I-70126 Bari, Italy; antonella.pasqualone@uniba.it (A.P.); giacomo.squeo@uniba.it (G.S.); carmine.summo@uniba.it (C.S.); 2Center of Food and Fermentation Technologies, Akadeemia tee 15a, 12618 Tallinn, Estonia; aleksei@tftak.eu (A.K.); helen@tftak.eu (H.V.); martti@tftak.eu (M.T.); mariliis.tammik@tftak.eu (M.-L.T.); 3School of Business and Governance, Department of Business Administration, Tallinn University of Technology, Ehitajate tee 5, 19086 Tallinn, Estonia; 4School of Science, Department of Chemistry and Biotechnology, Tallinn University of Technology, Ehitajate tee 5, 19086 Tallinn, Estonia

**Keywords:** extrusion cooking, extruder responses, dry fractionation, pea protein, plant-based meat analogues, sustainability, functional properties, sensory analysis

## Abstract

Pea protein dry-fractionated (P_DF_), pea protein isolated (P_Is_), soy protein isolated (S_Is_) and oat protein (O_P_) were combined in four mixes (P_DF__O_P_, P_Is__O_P_, P_DF__P_Is__O_P_, S_Is__O_P_) and extruded to produce meat analogues. The ingredients strongly influenced the process conditions and the use of P_DF_ required higher specific mechanical energy and screw speed to create fibrous texture compared to P_Is_ and S_Is_. P_DF_ can be conveniently used to produce meat analogues with a protein content of 55 g 100 g^−1^, which is exploitable in meat-alternatives formulation. P_DF_-based meat analogues showed lower hardness (13.55–18.33 N) than those produced from P_Is_ and S_Is_ (nearly 27 N), probably due to a more porous structure given by the natural presence of carbohydrates in the dry-fractionated ingredient. P_DF__O_P_ and P_Is__P_DF__O_P_ showed a significantly lower water absorption capacity than P_Is_ O_P_ and S_Is__O_P_, whereas pea-based extrudates showed high oil absorption capacity, which could be convenient to facilitate the inclusion of oil and fat in the final formulation. The sensory evaluation highlighted an intense odor and taste profile of P_DF__O_P_, whereas the extrudates produced by protein isolates had more neutral sensory characteristics. Overall, the use of dry-fractionated protein supports the strategies to efficiently produce clean-labeled and sustainable plant-based meat analogues.

## 1. Introduction

The world population is expected to reach 9.7 billion by 2050 [1], causing an increase in protein demand. Meat and dairy products are the primary protein source for the most of population of developed countries, with rising average consumption at a global level, but they require a considerable amount of resources such as arable lands, pastures, feed for livestock and water [2]. This leads to concerns regarding the affordability of feeding the world without damaging the environment [3]. In this scenario, plant-based food can efficiently contribute to satisfying the food demand in a sustainable and nutritionally balanced way. Plant-based protein alternatives are already established and available on the market and are mainly derived from soy (e.g., tofu and tempeh) or wheat gluten (e.g., seitan). However, a further step was carried out by developing meat analogues, which are characterized by the realistic imitation of meat structure and its functionality [4]. Textural and sensorial properties indeed are the key targets for the consumers, which require specific product characteristics [4]. Meat analogues are principally constituted by texturized vegetable proteins, i.e., a fibrous matrix of vegetable protein, imitating the fibrillar structure of meat muscle [5]. Structuring techniques of vegetable proteins were widely investigated in recent years [3,4,6]. The most used technology to produce meat analogues is the high-moisture extrusion cooking by a twin-screw extruder. This process produces big pieces of texturized protein, similar to muscle meat [7], useful to formulate chunk-type meat analogues [8]. However, meat analogues already available on the market principally include burgers and patties [9], for which a low-moisture extrusion cooking could be more convenient, because the derived texturized protein is a small pellet with a fibrous texture, and it can be easily rehydrated and mixed with the ingredients necessary for the production of finished meat analogues.

The raw materials for texturization are typically protein isolates derived from soy [10] and pea [7], but interest is rising for other protein sources, such as lupin [11], hemp [12], mung bean, and wheat gluten [13]. Recently, a pea–oat protein blend has been utilized to produce extruded meat analogues characterized by good amounts of essential amino acids and nutritionally valuable composition [14]. Moreover, extrusion cooking significantly reduces the antinutritional factors contained in pulses, enhancing the nutritional value [15]. Indeed, legumes contain phytic acids, tannins and inhibitors of digestive enzymes, that reduce their nutritional value and their acceptability [16]. As recently reviewed [15], the extrusion process is capable of reducing heat-labile compounds such as the phytic acid, trypsin inhibitor and tannins, due to the combination of the thermal treatment and the mechanical shear of the screw speed. This also leads to an increase in protein and starch digestibility, exploitable to produce ready to eat products of particular importance for both developing and developed countries.

Nonetheless, protein isolates are obtained by wet extraction, which requires a high quantity of energy and chemicals [17] placing the production of meat analogues actually in conflict with the purpose of enhancing the sustainability of food production. Dry fractionation is a sustainable alternative to the common wet extraction, but it produces protein concentrates of lower purity than protein isolates [18] and characterized by high complexity due to the presence of carbohydrates, minerals, bioactive compounds, and volatile molecules responsible for the typical legume flavor. Dry fractionation techniques mainly involve the use of air classification but also techniques under development, such as the tribo-electrostatic separation, can be effectively used [19,20]. Since this process does not involve the use of water and/or heating stages, the native state of the protein is preserved, resulting in a better functional performance in terms of protein solubility, foaming capacity, emulsion stability [20] and gelling properties [21], compared to the same proteins produced with wet extraction.

For this reason, future challenges involve the ability to use complex ingredients such as dry-fractionated proteins in order to reduce the environmental impact, the costs of productions [18], and the list of ingredients in the product, encouraging the preparation of low-processed food, with consequent benefits for both environment and human health, as recently expressed by Monteiro et al. [22]. Therefore, from the perspective of using few ingredients to produce clean-labelled and low-processed products, the use of dry-fractionated proteins could be a more efficient and sustainable strategy, because they already contain nutritional and functional compounds, such as carbohydrates and lipids.

So far, to the best of our knowledge, the use of dry-fractionated proteins as an ingredient for meat analogues has been just theorized in recent years [17], with few practical applications. For example, a recent study evaluated the possibility to use a dry heating method to improve the physicochemical properties of faba bean protein concentrate, making it suitable for meat replacers applications [23], whereas dry-fractionated pea and faba bean protein concentrates are currently used to produce snacks and texturized vegetable proteins (Organic Plant Protein A/S, Silkeborg, Denmark), confirming a rising trend. Therefore, in this research, we produced meat analogues by a low-moisture extrusion cooking from dry fractionated pea and oat protein with the aim to set up the extrusion cooking process for dry fractionated protein and to characterize the meat analogues for their physicochemical, functional, and sensory properties.

## 2. Materials and Methods

### 2.1. Materials

Pea protein isolates—P_Is_—(Careflour pea protein 80, Caremoli s.p.a, Monza, Italy) and defatted soy protein isolates—S_Is_—(Shandong Yuxin Bio-Tech Co., Ltd., Shandong, China), were chosen as the control because they are the most common ingredients for the preparation of commercial meat analogues [9]. Oat protein—O_P_—was produced by a combination of dry and wet milling (PrOatein, Lantmännen Oats, Kimstad, Sweden). In particular, the bran separated by dry milling was subjected to a wet milling process assisted by enzymes to separate a protein fraction that was finally spray-dried. Dry-fractionated pea protein—P_DF_—was kindly provided by InnovaProt s.r.l. (Gravina in Puglia, Italy) and it was produced by a plant of Separ Micro System sas (Flero, Brescia, Italy). The flour was first micronized twice in a KMX-300 micronizer equipped with a rotor operating at a peripheral speed of 157 m s^−1^. The disentanglement of starch and protein bodies was due to the mechanical impact against stator and rotor serrated surfaces and by turbulent multiple impacts between particles. The resulting flours were then air classified in an SX-100 apparatus, consisting of a turbo-separator and in a cyclone. The airflow is driven by an aspirating pump modulated by an inlet restriction-valve set to 270 in order to obtain a coarse fraction rich in starch (collected in the turbo-separator) and the fine fraction rich in protein (collected in the cyclone and used for this study).

### 2.2. Extrusion Process

The low-moisture extrusion process was carried out by the KETSE 20/40 twin-screw extruder (Brabender GmbH, Duisburg, Germany). The screw configuration was the same as that reported in Kaleda et al. [14], set to impart high shear. A total of four protein mixes where used for the production of meat analogues: (I) P_DF__O_P_: (70:30 *w*/*w*); (II) P_Is__O_P_: (70:30 *w*/*w*); (III) P_DF__P_Is__O_P_: (35:35:30 *w*/*w*/*w*); (IV) S_is__O_P_: (70:30 *w*/*w*). The protein mixes were chosen considering that combining pea protein with 30% of oat protein led to the production of stable textured proteins with important nutritional advantages related to the amino acid compensation [14]. The feeder was calibrated for each protein mix as well as the water pump. Extrusion conditions for P_Is__O_P_ were chosen according to Kaleda et al. [14], whereas for the other trials, we carried out preliminary tests, evaluating the visual appearance of the products (fibrous texture). The extrusion condition for each protein mix is reported in Table 1. The extruder was equipped with a short die of 2 mm in diameter. The extrudates were collected and subsequently dried for 2 h at 40 °C until water activity reached below 0.6 and stored at −18 °C until the analysis. The extrusion process was replicated on two different days. The specific mechanical energy (SME) was calculated with the formula reported by Fang et al. [24]:SME (kJ/kg) = (2π × rpm × T)/MFR(1) where T is the torque of the motor (Nm) calculated from the % of drive load, considering the maximum torque of the extruder as 80 Nm (for the calculation one minute was considered, with 60 measurements per minute); and MFR is the mass flow rate (g min^−1^), weighing the amount of extrudate produced within 2 min.

### 2.3. Proximate Composition of Protein Powders and Meat Analogues

The protein content (total nitrogen × 6.25), ash, and moisture content were determined according to the AOAC methods 979.09, 923.03, and 925.10, respectively [25]. The lipid content was determined by a Soxhlet apparatus, using diethyl ether (Merck KGaA, Darmstadt, Germany) as an extracting solvent (AOAC Method 945.38F [25]). Total carbohydrates content was determined as difference subtracting the protein, ash, moisture, and lipids contents from 100.

### 2.4. Texture Profile Analysis

Texture profile analysis (TPA) was carried out by a TA.XT2i texture analyzer (Stable Micro Systems Ltd., Godalming, UK) equipped with a cylindrical probe of 36 mm diameter. The conditions of the analysis followed a previously reported method [14], with slight modifications. Meat analogues were prepared for the analysis by rehydrating with tap water at 20 °C until the inner part showed no trace of dry material and cut to the length of 20 mm in order to have homogeneous pieces of extrudates. The diameter of the samples was about 5 mm. The rehydrated meat analogues are showed in Appendix A. A double compression cycle was performed at 1 mm s^−1^ until a recorded deformation of 75%. Fifteen replicates for each sample were considered. The following parameters were evaluated: Hardness: the maximum force recorded during the first compression; Cohesiveness: the area of work during the second compression divided by the area of work during the first compression; Springiness: the distance recorded during the second compression divided by the distance of the first compression; Chewiness: Hardness × Cohesiveness × Springiness.

### 2.5. Physicochemical and Functional Properties

Bulk density, water absorption (WAC), and oil absorption (OAC) capacities were determined for protein concentrates and ground meat analogues, according to Summo et al. [26].

The rehydration ratio was determined with whole meat analogues according to Tehrani et al. [10] with few modifications. Amounts of 5 g of meat analogues were rehydrated with 50 mL of water at 20 °C for 1 h. The rehydration ratio (RR%) was expressed in percentage, according to the following equation:RR% = (M_2_ − M_1_/M_1_) × 100(2)
where M_2_ is the weight of the sample after rehydration and M_1_ is the weight of the dried sample.

The specific volume of whole meat analogues was determined by rapeseed displacement AACC method 10-05.01 [27].

### 2.6. Sensory Analysis

Seven panelists, 6 women and 1 man (average age of 28 years) belonged to the expert sensory panel of Center of Food and Fermentation Technologies (Tallinn, Estonia), were selected based on their previous experience and training in sensory analysis (including assessing extruded protein products) and were informed of the study aims. A single training session was carried out with the panelists in order to identify the descriptors that best fitted the products in terms of appearance, odor, taste, and texture, but also to identify the ranges as well as setting the scale anchors of each descriptor. Verbal anchor points were used to calibrate the panel’s understanding of the scale. Physical references were used for all the attributes, where plant-based extruded samples had to be compared with meat. Panel performance was validated using PanelCheck (version 1.4.1—Nofima Mat, Breivika, Ås, Norway). The list of the sensory attributes, definitions, and scale is reported in Table 2. With respect to the textural parameters, the chewiness was evaluated by chewing the sample until it could be swallowed; the cohesiveness was evaluated by compressing the sample between teeth (one or more times) and focusing on deformation before rupturing; moisture and graininess were evaluated by masticating the sample with 5 chews. Panelists evaluated the samples on an 11-point structured scale ranging from 0 to 10. It was established that the scale had enough scale-points to distinguish the products, also to prevent the avoidance of scale-end anchors. For the descriptive sensory analysis, amounts of 5 g of rehydrated meat analogues (according to the times identified for the TPA) were put in a glass container, codified by a three-digit alphanumeric code, and then served to each panelist followed by Williams’ Latin square design. Sensory analysis was replicated on two different sessions. Each panelist carried out the sensory evaluation in a sensory room according to the ISO 8589:2007.

### 2.7. Statistical Analysis

The data were subjected to one-way ANOVA followed by Tukey’s HSD (honestly significant difference) test. Significant differences were determined at *p* ≤ 0.05 by Minitab 17 Statistical Software (Minitab, Inc., State College, PA, USA).

## 3. Results

### 3.1. Characteristics of the Ingredients

The chemical composition and functional properties of the raw materials used for the production of the meat analogues are reported in Table 3. Both protein isolates showed a protein content near 90 g 100 g^−1^. By contrast, dry-fractionated pea protein showed lower protein content, near 55 g 100 g^−1^ and this is in accordance with previous studies carried out on dry-fractionation by air classification of pulses flour [17]. Similar protein content was found in O_P_. P_DF_ and S_Is_ showed very similar ash content and these data corroborate other studies reporting the proximate composition of dry-fractionated pea protein [28] and soy isolates [29] whereas ash was lower in oat protein. The lipid content of S_Is_ was nearly negligible whereas it was considerably high in oat protein (16.85 g 100 g^−1^). Carbohydrates content varied based on the protein extraction method. Dry-fractionated proteins generally have a higher concentration of carbohydrates, whereas they are more efficiently separated during protein isolation [17]. Both carbohydrates and lipid content have proven to have a significant effect on texture formation during the extrusion process [6].

Considering the functional properties, soy protein demonstrated high functionality with the highest water absorption capacity, more than six times higher than P_DF_, which showed the lowest capacity to bind water, whereas O_P_ and P_Is_ showed similar WAC. Considering the same species (dry fractionated pea and isolated pea), it is interesting to highlight the differences between P_DF_ and P_Is_ in terms of WAC (lower in P_DF_ than in P_Is_). This is the result of protein denaturation that occurred during protein isolation [21] and is responsible for the rearrangement of hydrophilic groups of protein [30]. The WAC is also related to the presence of carbohydrates but also to the presence of fibers [31,32]. The WAC of coarse fractions obtained by air classification of pulse flours is indeed higher than the WAC of the fine fractions [32] and this is in line with our results. Oil absorption capacity showed slight but significant differences among the proteins, with the higher values displayed by S_Is_ followed by P_DF_. In the review by Lam et al. [30], it is reported that the functionality of the proteins depends on the extraction method. WAC and OAC are important functional properties in food formulation, giving information about the propensity of the protein to being mixed with other ingredients such as water or lipids.

### 3.2. Process Conditions and Extruder Responses

The extrusion process was slightly different among the four mixtures investigated, as reported in Table 1. In particular, when the P_DF_ was used, the process required more shear energy to obtain the texturization. Indeed, the screw speed for the P_Is__O_P_ mix was set to 600 rpm whereas it needed to be increased to 800 rpm for mixes containing the P_DF_. The temperature profile of the extruder zones, instead, was suitable for all the pea/oat mixes and remained constant. By contrast, soy protein required dramatically milder conditions in terms of screw speed and temperatures. Specifically, the S_Is__O_P_ mix needed 225 rpm and a decrease of 15 °C in the last three heating zones of the extruder to form a consistent pellet with a fibrous texture. In our preliminary tests, we found that higher screw speed and temperature led to the expansion of the product with the loss of the desirable meat-like texture.

It seems that with the same species, the process needs to be slightly calibrated by little regulation of screw speed and moisture (according to WAC). The amount of water required for the process was different, ranging from 35 g 100 g^−1^ of the S_Is__O_P_ mix to 20 g 100 g^−1^ of the P_DF__O_P_ mix, due to the different WAC of ingredients (Table 3). Indeed, soy protein showed the highest WAC, whereas P_DF_ showed the lowest capacity to bind water. Moisture content and process temperature are critical factors for the texture formation in meat analogues [6,8], therefore, the functional properties of proteins and other ingredients should be carefully considered in order to select the best process conditions to produce meat analogues. The process conditions are therefore related to a wide range of variables including the nature of the ingredient and its proximate composition. For instance, it is reported that substituting the soy protein with wheat gluten, caused a significant decrease in SME, while maintaining constant the protein content of the mixes [33]. Considering the same protein ingredient, previous studies reported that increasing the moisture content, therefore reducing the protein level of a lupin [11] and soy [29] protein isolate, led to a reduction of SME and to a change of the product characteristics. Therefore, each process needs to be calibrated according to the protein ingredient and the desired characteristics of the product.

Specific mechanical energy (SME) identifies the quantity of work (kJ) per output of the product (kg) and it characterizes the mechanical treatment of the material during extrusion [5,24]. Moreover, SME is an extruder response that can easily be used to compare and optimize extrusion processes involving different ingredients [5,24]. SME is directly related to the apparent viscosity of the melt, shear rate, and residence time in the extruder [34]. In our study, SME significantly varied among the four mixes, and it showed the highest value in P_DF__O_P_, followed by P_Is__P_DF__O_P_, whereas SME for the S_Is__O_P_ mix was the lowest. Fang et al. [24] previously reported that high SME led to a decrease in the degree of texturization for soy protein isolates, and this could explain the lowest SME required for the S_Is__O_P_ mix. Moreover, moisture content has a lubricant effect [29], therefore the higher SME of P_DF_ mixes could be related to a combination of the lower moisture used and the higher screw speed. Additionally, lipids have a lubricant effect in the extrusion process and a plasticizer role [6], however, in this case, lipids are bound into the protein powders and not added to the mix, therefore, it is reasonable to assume that their effect is negligible compared to the moisture content, which is directly added into the extruder. High torque variability during the extrusion process leads to a considerable high variation of SME that gives unstable and not uniform products at the die exit [5]. In our trials, the extruder responses were constant, therefore the process can be considered stable for all the mixes.

### 3.3. Chemical Composition of Meat Analogues

The proximate composition of meat analogues is reported in Table 4. Meat analogues produced with P_DF_ showed lower protein content than those produced by pea and soy protein isolates, according to the proximate composition of ingredients used in the formulation. However, the P_DF__O_P_ mix reached a protein content of 55 g 100 g^−1^ on dry matter, which is still suitable for the formulation of meat analogues products. Indeed, according to a previous investigation, the protein content of imitative burgers ranged between 17.7 and 25.0 g 100 g^−1^ [9]. In meat analogues, the protein ingredient used in the formulation is “diluted” by the addition of several ingredients to reach both a balanced nutritional composition, similar to meat products, and the desired textural properties. This is a cause of concern regarding the nutritional quality, the costs, and consumer confidence [8].

The lipid content of extrudates (Table 4) was essentially determined by the oat proteins and the dry fractionated pea proteins which contained a high quantity of fat (Table 3). Meat analogues produced from the dry-fractionated pea protein showed higher lipid content than those produced from pea isolate. When present in the range 2–10% of the powder mix [6], lipids have a positive role in the extrusion process due to a lubricant effect in the mixing zones of an extruder and the formation of lipid–protein and lipid–starch complexes in the melting zone. Therefore, the presence of residue lipids in both dry-fractionated pea and oat proteins could improve the extrusion process.

P_DF__O_P_ was characterized by the highest carbohydrate content, followed by P_Is__P_DF__O_P_. This was due to the different protein extraction methods, indeed, no significant differences were found among the meat analogues produced from protein isolates, naturally lacking in carbohydrates [21]. Although the use of a small quantity of starch in the extrusion process is favorable to the texture formation of meat analogues [35], generally, high content of carbohydrates is undesirable, due to the tendency of starch to absorb water, swell, and expand at the die exit [6], causing the loss of the fibrous texture. In our experiments, maintaining low moisture content in trials containing dry-fractionated protein could have helped the inhibition of the gelatinization process, limiting the expansion of the extrudates, together with the effect of the low content of carbohydrates, and the presence of lipids [15].

Finally, ash content varied little among the trials, although we detected significant differences among P_Is__O_P_ and P_Is__P_DF__O_P_ which showed lower ash contents than P_DF__O_P_ and S_Is__O_P_ meat analogues. These differences can be attributed to the differences in the proximate composition of the raw materials, in which P_DF_ and S_Is_ showed the highest ash content.

### 3.4. TPA, Physicochemical and Functional Properties

Results of the texture profile analysis of rehydrated meat analogues are reported in Table 5. The rehydration time was different for the blends as follows: P_DF__O_P_, P_Is__O_P_, P_DF__P_Is__O_P_, S_is__O_P_ for 10, 35, 20, and 55 min respectively. It is possible to highlight the significantly higher hardness (nearly 27 N) of extrudates produced from protein isolates than those produced by dry-fractionated proteins, with P_DF__O_P_ requiring half of the force to be compressed. The mix P_Is__P_DF__O_P_, being a combination of P_DF_ and P_Is_, showed an intermediate hardness. The use of dry-fractionated proteins affected also the cohesiveness, springiness, and chewiness, clearly distinguished into two distinct groups, of which the first was constituted by P_Is__P_DF__O_P_ and P_DF__O_P_ both characterized by the lowest textural indices. By contrast, P_Is__O_P_ and S_Is__O_P_ showed the highest values for cohesiveness, springiness, and chewiness, and elastic behavior indicating that these products need more energy to be chewed. The results can be explained by the different chemical composition of the ingredients. P_DF_ indeed showed the highest content of carbohydrates, which tend to expand during extrusion, creating a porous structure. The mechanical strength of extrudates is reduced if the product is expanded [36]. This can also justify a negative and significant correlation between total carbohydrates and hardness (Pearson’s r = −0.992; *p* = 0.008), and total carbohydrates and chewiness (r = −0.963, *p* = 0.037). Furthermore, it is probable that the high protein content of meat analogues produced from protein isolates led to the formation of a dense fibrous network, due to covalent isopeptide cross-linking [35] and more sulfur-links can be formed for creating the texture in the extruded product, leading to harder texture [12]. To support this hypothesis, positive and significant correlations were found between protein content and hardness (r = 0.973; *p* = 0.027) and chewiness (r = 0.980; *p* = 0.020).

The physicochemical and functional properties of the extrudates are also reported in Table 5. The specific volume of P_DF__O_P_ showed the highest absolute value, but it was statistically similar to P_Is__O_P_, indicating that the product was more expanded and less dense than S_Is__O_P_. This is also confirmed by the determination of bulk density which was also significantly lower in P_DF__O_P_ than in other formulations. However, the expansion of P_DF__O_P_ did not cause the loss of the fibrous texture and this result could be also ascribed to the different compositions of the proteins. In particular, it could be that soy protein tolerates expansion phenomena less than pea protein (i.e., rapid loss of fibrosity), probably due to the lower presence of lipids. It is reported that protein–lipid and carbohydrate–lipid interactions indeed, better stabilize the fibrous structure [6].

Water absorption capacity (WAC) is related to the presence of polar molecules capable of binding water [7,31]. WAC of extrudates significantly varied as influenced by the ingredients. P_DF_-based meat analogues showed the lowest capacity to bind water, whereas the use of P_Is_ allowed slightly, but significantly higher WAC. S_Is__O_P_ showed the highest WAC, following the same behavior observed in the raw materials (Table 3). Moreover, the results of WAC are confirmed by a previous study [13], which reported higher WAC for soy-based meat analogues compared to those produced by pea protein. From a practical point of view, higher WAC can help to enhance the water retention of meat analogues [37] leading to more appreciated products. No significant differences were identified for the rehydration ratio of pea-based meat analogues, whereas S_Is__O_P_ extrudates showed the highest rehydration ratio.

Oil absorption capacity (OAC) is associated with capillarity interactions [38], therefore, a more porous and expanded structure means a higher capacity to entrap oil. In our study, OAC showed the opposite trend found for WAC, evidencing the significantly lower capacity of S_Is__O_P_ to absorb oil than the other meat analogues. This was in accordance with Samard and Ryu [13]. Overall, it is possible to highlight that the use of dry-fractionated pea did not cause large changes in the functionality of the products in comparison to those produced with pea protein isolate, demonstrating that after calibration of the process parameters, such ingredients are suitable for the production of meat analogues with good texture and functional properties that are useful for further processing.

### 3.5. Sensory Analysis

The results of the sensory analysis are reported in Table 6. The color of meat analogues was clearly distinguished by panelists, according to the anchors reported in Table 2.

The use of protein isolates conferred a grey tone to meat analogues, whereas P_DF_ led to products with brown-greenish colors. Considering the pea proteins, the use of P_DF_ did not cause significant variations in the perception of the fibrous texture of meat analogues, whereas it was significantly lower in S_Is__O_P_. The use of dry-fractionated protein gave the highest overall odor intensity than the isolated ones and this was likely due to the combination of the strongest perception of cereals, sweetness, and meat-like odor notes. Therefore, the use of P_DF_ gave a more characteristic odor profile compared to both P_Is_ and S_Is_, the latter showing neutral appearance and odor. Indeed, as hypothesized above, a more intense Maillard reaction could have led to a more intense color and sensory profile of P_DF__O_P_. Moreover, dry-fractionated ingredients still contain volatile compounds responsible for the legume odor [17] which is not always appreciated by consumers. Generally, to mask the undesired flavors, spices and herbs are used as flavoring agents [9]. In our study, the incorporation of the 30% of oat protein might have contributed to smooth the odor intensity, improving the sensory characteristics, together with the favorable effect of the extrusion process [15]. Taste of meat analogues followed the same trend of odor notes, as the P_DF__O_P_ showed the greatest overall intensity together with the most intense cereal notes, umami, bitterness, and aftertaste. Overall, the use of pea protein isolates smoothed the sensory characteristics, and this effect was surprisingly evident in the P_Is__P_DF__O_P_ mix, of which the sensory characteristics were similar to P_Is__O_P_. Particularly interesting was the trend of cereals attribute, which significantly varied among the trials even considering that oat content was the same. It is possible that the presence of dry fractionated pea protein had a synergistic effect on cereal perception, masking the legume flavor, and enhancing the cereal notes. Additionally, the higher temperature used in the process could have contributed to the increase in odor and taste intensity. Undesirable attributes associated often with plant-based proteins are bitterness and astringency, which can be caused by the presence of antinutritional factors such as saponins, or phenolic compounds such as tannins and catechins [39,40]. Dry-fractionated proteins generally have higher antinutritional factors than protein isolates [21], which can also explain the bitter taste was perceived stronger in the P_DF__O_P_ mix than in the other trials. A strong bitter taste can hamper the inclusion of plant-based meat substitutes into the diet [41]. However, this aspect is less important for vegetarian consumers, as studies have shown that they generally have lower sensitivity to bitter taste [41]. Off-taste was perceived only in S_Is__O_P_, however, it was in very low intensity described as soapiness by some assessors. Texture evaluation by sensory analysis was following the results of instrumental analysis, confirming that the use of P_DF_ contributed to the formation of a softer product compared to both pea and soy protein isolates. This aspect should be considered if it is planned to use the extrudates from protein isolates in burger-type products, since harder pieces could be difficult to bind together to obtain a homogeneous texture. Interestingly, moisture perceived in the mouth was significantly higher in products containing P_DF_. Considering the WAC values, this could be a contrasting result, however, it is probable that during chewing, the porous structure of P_DF__O_P_ promoted the release of loosely held water.

It should be noted that the desired attributes of the meat substitutes depend largely on the end product goal as well as the target consumers. For example, some consumers are looking for meat taste in meat substitutes and others may prefer a taste that does not resemble meat [42]. For the first group of consumers, the sensory characteristics should be modulated in order to minimize the impact of legume and cereal notes, which means using a combination of protein isolates and dry fractionated protein. In plant-based meat alternatives, the low juiciness of cooked product is a common drawback, making the use of water-binding agents necessary in the formulation [8]. In this perspective, the higher moisture perceived in the sample produced by dry-fractionated protein can be a positive attribute. Since the protein extrudates produced in this study are semi-finished products for the formulation of a meat substitute, further consumer studies would be needed to select desired attributes for the product target. Overall, the sensory characteristics of extrudates produced from dry-fractionated pea protein support the idea of using these ingredients for the production of meat analogues.

## 4. Conclusions

Low-moisture extrusion cooking was successfully used for the production of texturized protein by using different protein mixes. Process conditions were strongly dependent on the raw materials and the use of dry-fractionated protein required a more intense SME (i.e., screw speed) to create fibrous texture compared to protein isolates. Overall, the extruder responses were constant within the process, giving stable production of texturized proteins. Results proved that dry fractionated protein concentrates can be conveniently used for the production of meat analogues in spite of lower protein content than protein isolates. The P_DF__O_P_ mix reached protein content of 55 g 100 g^−1^ on a dry matter basis, which is suitable for the formulation of meat analogue products, because it already contains a considerable amount of starch, lipids, and minerals which have nutritional and functional properties and which are minimally present in a protein isolates. P_DF_-based meat analogues showed lower hardness than those produced from pea and soy protein isolates, probably because of the presence of a more porous structure. P_DF__O_P_ and P_Is__P_DF__O_P_ showed a significantly lower WAC than P_Is__O_P_ and S_Is__O_P_, whereas pea-based extrudates were characterized by high OAC, which could be convenient to facilitate the inclusion of oil and fat in the formulation of the final product. The sensory evaluation highlighted an intense odor and taste profile of P_DF__O_P_, whereas the extrudates produced by protein isolates had more neutral sensory characteristics. Further research is needed to modulate and calibrate each of the sensory attributes for specific purposes and target consumers during the formulation of meat substitutes. Overall, we believe that the use of dry-fractionated protein is an efficient and sustainable strategy from the perspective of using few ingredients to produce clean-labelled plant-based meat analogues and this research is a step forward in this field.

## Figures and Tables

**Table 1 foods-09-01754-t001:** Process conditions and extruder responses for the four mixes. T_1–6_ temperature in the heating zones of the extruder. SME: specific mechanical energy; P_DF__O_P_: dry-fractionated pea protein and oat protein; P_Is__O_P_: pea protein isolates and oat protein; P_DF__P_Is__O_P_: dry-fractionated pea protein, pea protein isolate, and oat protein; S_Is__O_P_: soy isolates and oat protein.

Sample	Screw Speed (rpm)	T_1_ (°C)	T_2_ (°C)	T_3_ (°C)	T_4_ (°C)	T_5_ (°C)	T_6_ (°C)	Moisture (g 100 g^−1^)	Protein Content (g 100 g^−1^) ^§^	Lipid Content (g 100 g^−1^) ^§^	P * (bar)	Torque * (Nm)	Mass Flow Rate ** (g min^−1^)	SME * (kJ kg^−1^)
P_DF__O_P_ (70:30)	800	40	70	130	150	140	140	20	54.7	8.2	38.05 ± 2.50 ^a^	24.09 ± 2.57 ^a^	94.15 ± 5.28 ^a^	1286.01 ± 114.22 ^a^
P_Is__O_P_ (70:30)	600	40	70	130	150	140	140	30	77.5	7.1	16.28 ± 0.52 ^c^	14.55 ± 1.15 ^c^	71.00 ± 0.55 ^c^	772.60 ± 60.90 ^c^
P_Is__P_DF__O_P_ (35:35:30)	800	40	70	130	150	140	140	25	66.1	7.7	18.48 ± 3.72 ^b^	17.30 ± 2.13 ^b^	81.86 ± 6.88 ^b^	1075.77 ± 145.43 ^b^
S_Is__O_P_ (70:30)	225	40	70	125	135	125	125	35	76.9	5.3	18.14 ± 2.33 ^b^	12.84 ± 1.42 ^d^	71.50 ± 8.72 ^c^	226.94 ± 25.12 ^d^

Different letters in the same column mean significant differences at *p* < 0.05. ^§^ calculated from the protein content of the ingredients (Table 3). * data recorded within a minute, during the sample collection, with 60 measurements per minute (repeated for the two replicates of the process); ** *n* = 3.

**Table 2 foods-09-01754-t002:** Sensory attributes, definition, and scale anchors for meat analogues (scale 0–10).

	Attribute	Definition	Scale Anchors
Appearance	Color	Perceived color tone	0—brown; 3.5—yellow; 6.5—green; 10—grey
	Fibrousness	Number of fibers perceived in the sample	0—not fibrous; 10—very fibrous
Odor	Overall intensity *	Overall odor intensity of the sample	0—not perceived odor; 3—boiled chicken meat (overall); 7—boiled pork meat (overall); 10—very intense odor
	Meat-like **	Perceived similarities with meat	0— not resembling meat; 10—very similar to meat (boiled pork/chicken)
	Cereals	Association with cereals	0—none; 2—very mild; 4—mild; 6—moderate; 8—intense; 10—very intense
	Legumes	Association with legumes	
	Sweetness	Association with caramel/sugar	
	Off-odor intensity	Non-characteristic odors (chemical, rancid, metallic, etc.)	
Taste	Overall intensity *	Overall taste intensity of the sample	0—not perceived taste; 10—very intense taste (5—boiled pork/chicken)
	Cereals	Association with cereals	0—none; 2—very mild; 4—mild; 6—moderate; 8—intense; 10—very intense
	Legumes	Association with legumes	
	Saltiness	Association with sodium chloride	
	Sweetness	Association with sucrose	
	Umami	Association with glutamate	
	Bitterness	Association with caffeine	
	Astringent	Puckering sensation in mouth/tongue	
	Off-taste intensity	Non-characteristic tastes (chemical, rancid, metallic, etc.)	
	Aftertaste intensity	Intensity 5 s after swallowing the sample	
Texture	Springiness	Rate to which the sample recovers to its initial condition after pressing it with fingers	0—not recovering; 3—recovers slightly (boiled pork); 10—recovers completely (gummy candy)
	Hardness	The force required to compress the sample using teeth	0—easily compressible; 8—hardly compressible (boiled pork); 10—not compressible
	Cohesiveness	The amount of sample that holds together during chewing rather than rupturing	0—sample completely ruptured (halva ^§^); 9—very cohesive (boiled pork); 10—sample holds together completely
	Chewiness	Effort required to chew the sample until it can be swallowed	0—no chews needed for masticating food (liquid-like); 8—moderately hard to chew (boiled pork); 10—requires lot of chews to masticate food
	Moisture	Amount of water in the sample released during 5 chews	0—completely dry; 1—very dry (halva ^§^); 10—very moist
	Graininess	Amount of particles released during 5 chews	0—no particles perceived; 5—moderately grainy (halva ^§^) 10—very grainy, gritty mouthfeel

^§^ Halva: very friable and dry sweet confection (dense dry paste made of nuts and sugar). * overall intensity represents the intensity of the whole odor profile regardless of the type of the odor. ** Finding similar nuances either to pork meat or chicken meat means already that the sample has meat-like odor.

**Table 3 foods-09-01754-t003:** Proximate composition (g 100 g^−1^ dry matter) and functional properties of the protein powders used as raw materials for the extrusion process. P_DF_: dry-fractionated pea protein; P_Is_: pea protein isolates; O_P_: oat protein; S: soy protein isolates.

	P_DF_	P_Is_	S_Is_	O_P_
Protein	55.65 ± 0.17 ^c^	88.21 ± 0.43 ^a^	87.27 ± 0.04 ^b^	52.56 ± 0.33 ^d^
Lipids	4.54 ± 0.37 ^b^	2.93 ± 0.17 ^c^	0.38 ± 0.01 ^d^	16.85 ± 0.15 ^a^
Carbohydrates	34.92 ± 0.45 ^a^	5.00 ± 0.64 ^d^	7.58 ± 0.07 ^c^	28.00 ± 0.44 ^b^
Ash	4.88 ± 0.25 ^a^	3.86 ± 0.04 ^b^	4.77 ± 0.04 ^a^	2.60 ± 0.04 ^c^
WAC (g water g^−1^)	0.93 ± 0.04 ^c^	2.82 ± 0.05 ^b^	6.46 ± 0.05 ^a^	2.41 ± 0.39 ^b^
OAC (g oil g^−1^)	1.32 ± 0.06 ^b^	1.11 ± 0.06 ^bc^	1.66 ± 0.08 ^a^	1.02 ± 0.12 ^c^
BD (g mL^−1^)	0.74 ± 0.00 ^b^	0.76 ± 0.00 ^a^	0.44 ± 0.01 ^d^	0.60 ± 0.01 ^c^

Data expressed on dry matter. Different letters in the same row mean significant differences at *p* < 0.05. *n* = 3. WAC: water absorption capacity; OAC: oil absorption capacity; BD: bulk density.

**Table 4 foods-09-01754-t004:** Proximate composition of meat analogues (g 100 g^−1^ d.m.). P_DF__O_P_: dry-fractionated pea protein and oat protein; P_Is__O_P_: pea protein isolates and oat protein; P_DF__P_Is__O_P_: dry-fractionated pea protein, pea protein isolate, and oat protein; S_Is__O_P_: soy isolates and oat protein.

	P_DF__O_P_	P_Is__O_P_	P_Is__P_DF__O_P_	S_Is__O_P_
Protein	55.59 ± 0.93 ^d^	75.66 ± 0.30 ^a^	63.33 ± 0.35 ^c^	75.63 ± 0.68 ^a^
Lipids	8.93 ± 0.82 ^a^	6.55 ± 0.31 ^b^	7.52 ± 1.25 ^ab^	4.26 ± 0.02 ^c^
Carbohydrates	31.40 ± 0.46 ^a^	14.58 ± 0.26 ^c^	25.70 ± 0.68 ^b^	15.56 ± 0.73 ^c^
Ash	4.07 ± 0.16 ^a^	3.21 ± 0.21 ^b^	3.45 ± 0.22 ^b^	4.25 ± 0.14 ^a^

Data expressed on dry matter. Different letters in the same row mean significant differences at *p* < 0.05. *n* = 4.

**Table 5 foods-09-01754-t005:** TPA and physicochemical and functional properties of meat analogues. P_DF__O_P_: dry-fractionated pea protein and oat protein; P_Is__O_P_: pea protein isolates and oat protein; P_DF__P_Is__O_P_: dry-fractionated pea protein, pea protein isolate, and oat protein; S_Is__O_P_: soy isolates and oat protein.

	P_DF__O_P_	P_Is__O_P_	P_Is__P_DF__O_P_	S_Is__O_P_
Hardness (N) *	13.55 ± 2.60 ^c^	27.90 ± 4.76 ^a^	18.33 ± 4.25 ^b^	27.33 ± 5.30 ^a^
Cohesiveness *	0.54 ± 0.04 ^b^	0.59 ± 0.02 ^a^	0.54 ± 0.02 ^b^	0.62 ± 0.03 ^a^
Springiness *	0.72 ± 0.08 ^b^	0.81 ± 0.07 ^a^	0.74 ± 0.05 ^b^	0.87 ± 0.07 ^a^
Chewiness (N) *	5.25 ± 1.16 ^b^	13.46 ± 2.72 ^a^	7.27 ± 1.59 ^b^	14.77 ± 3.95 ^a^
Specific Volume (mL g^−1^) **	4.06 ± 0.19 ^a^	3.76 ± 1.05 ^ab^	2.60 ± 0.30 ^bc^	2.05 ± 0.25 ^c^
BD (g mL^−1^) **	0.43 ± 0.02 ^d^	0.55 ± 0.01 ^c^	0.57 ± 0.00 ^b^	0.70 ± 0.01 ^a^
WAC (g water g^−1^) **	2.19 ± 0.07 ^c^	2.45 ± 0.11 ^b^	2.15 ± 0.09 ^c^	3.42 ± 0.10 ^a^
RR (%) **	223.31 ± 11.44 ^b^	210.96 ± 11.13 ^b^	205.13 ± 11.11 ^b^	257.74 ± 8.53 ^a^
OAC (g oil g^−1^) **	1.85 ± 0.29 ^a^	1.57 ± 0.19 ^a^	1.45 ± 0.17 ^ab^	1.06 ± 0.10 ^b^

Different letters in the same row mean significant differences at *p* < 0.05. * *n* = 15; ** *n* = 4. BD: bulk density; WAC: water absorption capacity; RR: rehydration ratio; OAC: oil absorption capacity.

**Table 6 foods-09-01754-t006:** Sensory analysis of meat analogues. P_DF__O_P_: dry-fractionated pea protein and oat protein; P_Is__O_P_: pea protein isolates and oat protein; P_DF__P_Is__O_P_: dry-fractionated pea protein, pea protein isolate, and oat protein; S_Is__O_P_: soy isolates and oat protein.

	Attribute	P_DF__O_P_	P_Is__O_P_	P_Is__P_DF__O_P_	S_Is__O_P_
Appearance	Color	1.29 ± 0.91 ^d^	8.00 ± 0.96 ^b^	6.00 ± 0.78 ^c^	9.14 ± 0.77 ^a^
	Fibrousness	5.79 ± 0.97 ^a^	5.86 ± 0.95 ^a^	6.64 ± 0.93 ^a^	3.79 ± 0.97 ^b^
Odor	Overall intensity *	8.07 ± 0.83 ^a^	5.64 ± 0.93 ^b^	5.29 ± 0.91 ^b^	4.29 ± 0.47 ^c^
	Meat-like **	3.50 ± 0.85 ^a^	1.29 ± 0.91 ^b^	1.07 ± 0.92 ^b^	0.00 ± 0.00 ^c^
	Cereals	6.64 ± 0.84 ^a^	5.21 ± 0.97 ^b^	5.14 ± 0.77 ^bc^	4.29 ± 0.99 ^c^
	Legumes	1.36 ± 1.74 ^a^	0.43 ± 0.76 ^ab^	0.64 ± 0.84 ^ab^	0.00 ± 0.00 ^b^
	Sweetness	4.71 ± 0.91 ^a^	2.79 ± 0.97 ^b^	2.79 ± 0.80 ^b^	2.71 ± 0.99 ^b^
	Off-odor intensity	0.00 ± 0.00	0.00 ± 0.00	0.00 ± 0.00	0.00 ± 0.00
Taste	Overall intensity *	5.71 ± 0.83 ^a^	3.64 ± 0.93 ^bc^	3.86 ± 0.95 ^b^	2.86 ± 0.86 ^c^
	Cereals	4.86 ± 0.95 ^a^	3.29 ± 0.91 ^b^	3.21 ± 0.80 ^b^	2.86 ± 1.03 ^b^
	Legumes	1.57 ± 1.50 ^a^	1.00 ± 1.18 ^ab^	1.07 ± 1.14 ^ab^	0.00 ± 0.00 ^b^
	Saltiness	1.71 ± 0.83 ^a^	1.29 ± 0.73 ^a^	1.43 ± 0.76 ^a^	1.14 ± 0.86 ^a^
	Sweetness	1.64 ± 0.84 ^a^	1.14 ± 0.36 ^a^	1.50 ± 0.94 ^a^	0.93 ± 0.73 ^a^
	Umami	1.14 ± 0.86 ^a^	0.00 ± 0.00 ^b^	0.00 ± 0.00 ^b^	0.00 ± 0.00 ^b^
	Bitterness	2.64 ± 0.84 ^a^	1.79 ± 0.80 ^b^	1.71 ± 0.47 ^b^	1.29 ± 0.47 ^b^
	Astringent	1.36 ± 0.63 ^a^	1.14 ± 0.86 ^a^	1.00 ± 0.78 ^a^	1.21 ± 0.43 ^a^
	Off-taste intensity	0.00 ± 0.00 ^b^	0.00 ± 0.00 ^b^	0.00 ± 0.00 ^b^	1.57 ± 0.76 ^a^
	Aftertaste intensity	3.93 ± 0.73 ^a^	2.43 ± 1.02 ^b^	2.43 ± 0.94 ^b^	2.64 ± 0.93 ^b^
Texture	Springiness	4.64 ± 1.01 ^b^	5.93 ± 0.92 ^a^	3.86 ± 0.95 ^b^	6.29 ± 0.99 ^a^
	Hardness	4.93 ± 0.92 ^b^	7.93 ± 1.00 ^a^	5.71 ± 0.91 ^b^	8.00 ± 0.88 ^a^
	Cohesiveness	6.64 ± 0.84 ^a^	7.07 ± 0.92 ^a^	6.64 ± 0.93 ^a^	7.00 ± 1.04 ^a^
	Chewiness	4.71 ± 0.99 ^c^	8.29 ± 0.91 ^a^	6.21 ± 0.89 ^b^	8.00 ± 0.88 ^a^
	Moisture	8.79 ± 0.80 ^a^	6.21 ± 0.80 ^b^	7.93 ± 0.92 ^a^	6.57 ± 0.94 ^b^
	Graininess	0.00 ± 0.00 ^a^	0.71 ± 0.91 ^a^	0.43 ± 0.76 ^a^	0.64 ± 1.01 ^a^

Different letters in the same row mean significant differences at *p* < 0.05. * Overall intensity represents the intensity of the whole odor profile regardless of the type of the odor. ** Finding similar nuances either to pork or chicken means already that the sample has meat-like odor.

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
