# Peer review of "Physicochemical and Sensorial Evaluation of Meat Analogues Produced from Dry-Fractionated Pea and Oat Proteins"

_foods, 2020, doi:10.3390/foods9121754_

Round 1

Reviewer 1 Report

The manuscript investigated the use of dry fractionated protein powders in production of meat analogues and assessed the physicochemical and sensorial properties of the resulting meat analogous compared to those produced by protein isolates through wet fractionation processes. The study is novel and interesting.

ABSTRACT: The abstract should stand alone. Some modification is required to revise abstract as below.

Line 16: Is oat protein also dry fractionated? If yes, please be consistent and use DF in the abbreviation so that readers will realize the oat protein is dry fractionated.

Line 17: what do the authors mean by stable in this sentence “A stable extrusion process was achieved with all the mixes”. Also what mixes you are referring to?

Line 17: what are raw materials??

Line 23: Why dry fractionated Pea protein has porous structures compared to isolates?

INTRODUCTION: The introduction is organized, but very general information is explained. This can be improved by incorporating more literatures. The authors need to explain more about type of the dry fractionation processes being recently developed for beans and cereals, focusing on their functional properties compared to protein isolates.

Line 37: what kind of resources?

Line 58: what are antinutritional factors? And how does extrusion reduce the antinutritional factors in pulses? This needs very brief elaboration with respect to literatures.

Line 64: Replace “e.g.” with “due to the presence of ……”.

Lines 67-68: how does the use of dry fractionated protein reduces the list of ingredients in the product? What do authors mean by “list of ingredients in the product”? Dry fractionated proteins have more ingredients compared to protein isolates.

Line 68: Why using dry fractionated protein powders in production of meat analogues is considered low-processed food?

Some information in lines 229-232 can be used here to clarify the case.

Line 71: Name and explain the practical applications with related references.

MATETRIALS:

Line 77: What kind of milling and dry fractionated process was being used to produce oat and pea protein powders. I understand that they are being purchased; however, the authors should be able to provide more detailed information from vendors. It is important to include the type of the dry fractionated protein powders tested in this study to produce meat analogues.

It is also important to include type of the milling used prior to the dry fractionation process.

Line 90-91: The extrusion process for each protein mix has to be mentioned.

Line 119: “grams” can be replaced by “g”.

Lines 129-131: Please modify this sentence “A training session was carried out with the panelists in order to identify the descriptors that best fitted the products in terms of appearance, odor, taste, and texture, identify the ranges and set the scale anchors of each descriptor.” You may use “A training session was carried out with the panelists not only to identify the descriptors that best fitted the products in terms of appearance, odor, taste, and texture, but also to identify the ranges as well as setting the scale anchors of each descriptor.”

Line 140: spell out HSD.

Table 1: Why some of the attributes do not have scale anchors? Table one is not clear and it is hard to distinguish the definition of attributes and the scales. The rationale behind the scales can be described as a footnote.

RESULTS AND DISCUSSION:

Line 148: what kind of dry fractionation?

Line 149: Oat grouts do not have that high (~50%) proteins. What kind of milling and dry fractionation was used?

Line 183 and line 1893: what authors mean by “species” here?

Line 183: This conclusion of “Therefore, it is probable that the species, rather than the extraction method (dry or wet fractionation), could have more influence on the extrusion parameters” need more elaboration and evidence!!

Line 203: what kind of processes??

Lines 201-203, the sentence has to be modified!

Lines 206-210: How does this statement relate to your results?? “Fang et al. [19] previously reported that high SME led to a decrease of the degree of texturization for soy protein  isolates, whereas Osen et al. [7] found that SME was affected by both the ingredients and the temperatures of the extruder because they had a consistent influence on the viscosity of the matrix inside the extruder.”

LINE 213: WHAT DO YOU MEAN BY “synonym of incorrect process conditions [5]”?

Lines 2016-2018: please revise the sentence.

Lines 229-232: This paragraph should be mentioned in the introduction to clarify the reason of using few ingredient and low-processed foods. This information basically belongs to the Introduction section.

Line 281 to 285: Please revises by breaking the whole sentence down to a few ones.

Reviewer 2 Report

The presented study reports the physicochemical and sensorial properties of extruded meat analogues produced using dry fractionated pea and oat protein in comparison to extruded meat analogues produced using either pea or soy protein isolate in combination with oat protein, as well as dry fractionated pea in combination with pea protein isolate and oat protein. These results are interesting especially for applications of these products in food technology. However, the following points can be considered in revising this manuscript:

Line 81: was produced a combination – correct to “was produced by a combination”

Line 105: 1000 – should be 100

Line 111-112: The rehydration time was different for the blends as follows: PDF_OP, PIs_OP, PDF_PIs_OP, Sis_OP for 10, 35, 20, and 55 minutes respectively – is this rather a result and should therefore be reported in the results section.

Line 138: refers to the product “halva” – I think it is difficult for the reader to understand what you are comparing to as this is not a generally well-known product. I would suggest that you include another reference product.

Line 153-154: have higher carbohydrates – change to “have higher concentrations of carbohydrates”

Line 166-167: I think the authors also need to discuss how the different contents of carbohydrates are expected to affect the WAC

Table 3: I will suggest to include the ratio of the ingredients in the table text or in the table.  

Regarding table 2 and table 3: I miss a clear overview of the proximate contents of the mixes before extrusion, as this would make an easier comparison – it could be either as a separate table, combined with table 2 or simply the total protein content of the combined ingredients added to table 3.

Table 3: I suggest to add the temperature unit to the table below the T1-T6, instead of having it in the table text.

Line 239: it would be good to have a clear and easy overview of the lipid content of the mixes – see also similar note about the protein content above (“Regarding table 2 and 3……”)

Table 5: please check that the last two values for OAC are indeed overlapping statistically

The authors discuss the many different attributes evaluated in table 6, but I miss a clear indication about which attributes are desired/wanted and which are undesired in order to create the best and most attractive products.  

Reviewer 3 Report

I found the study and the paper very interesting and well written.

Line 88-89: Is the ratio in the mixtures based on the protein content or the complete sample? Since the different samples has different protein contents and will therefore behave differently in the extrusion process. What is the moisture content of the raw materials and in the mixed samples? This is needed for the correct moisture contents for the extruder process.

Line 109: I cannot see any conditions being similar except the speed, in your texture analysis as compared to the reference your stated. 1. You are using TPA, the ref does not mention if they are using a single or double compression. 2. Probe diameters are different, 3. The compression distance is different 50% vs 75%. And in addition, I cannot see that their measurement is working as described, since they have a sample size of 1.5mm x1.5mm, which cannot be compressed to 50% during 5s with a speed of 1mm/s, so either there is a holding time or if it is a TPA, a resting time.

With a speed of 1mm/s and a compression distance to 75% of the sample height for a sample with a thickness of 2(?)mm diameter, the measurements are not always very accurate due to the mechanical movement i.e. the accuracy of the instrument and method.

Line 113: What was the diameter of your samples? Was the sample cylindrical shaped? If you have different diameters of the samples, and if they are cylindrical shaped, then the contact area (compression area) differs between the samples which might influence the results.

Line 123: How big were the pieces, were they solid enough for not being compressed when measured by rapeseed displacement method? How big was the method variation?

Line 132: Was the rehydration done taking in consideration the different times mentioned in Line 112 OR 1 hour as mentioned in Line 120?

Table 1: Which scale was used for the cereals/legumes/sweetness/saltiness/Off-taste intensity/after taste intensity? Please clarify in the table.

With chewiness/cohesiveness/moisture/graininess, was the evaluation done after a certain number of chews or it was up to the panelists to decide?

Line 173-218: How much of the differences in the process is related or depending on the different protein contents in your samples? Cannot see any discussion about the importance of the protein content in this section.

Line 210: Lipids also have a lubricant effect in extrusion. Is water or lipids more lubricant? And is that the explanation between the SME for PIS_OP vs SIS_OP?

Line 233: In materials it stated that the OP was both dry and wet milled, not only dry fractionated, so please clarify.

Table 3: Are the individual moisture contents taken in consideration when calculating the moisture content for the extrusion process?

Line 246: What was your moisture content in the extruded products?

Line 259: Is the protein isolates denser i.e. lower specific volume? The moisture content could also have an impact as well as if the sizes of the products are different. A small different in size for small samples can contribute to a large variation in the results.

Line 265-267: Do you think this is due to the higher level of proteins in these samples or a result of the other components?

Line 271-273: I agree, since due to higher protein levels, more sulfur-links can be formed for creating the texture in the extruded product.

Table 5: Please show how the Cohesiveness, Springiness and Chewiness are calculated? For instance, the Chewiness is Hardness*Cohesiveness*Springiness.

Line 310-348: I am missing a discussion regarding anti-nutrients in the sensory evaluation regarding taste and odor. Can it be that the ANF are contributing to the odor and flavor properties?

Reviewer 4 Report

The article is interesting because, as the authors state in the introduction section, there is no work on the use of dry-fractionated proteins. However, some point need to be addressed to clarify some aspects of the manuscript.

Materials and methods

What is the reason for choosing the mixtures tested? The mixtures always include a 30% of Op and variable amount of the dry-fractionated pea protein or other protein isolates. Why this amount?

It would very interesting to test PDf_Sis_Op also.

What is the reference formula? In other words, what characteristics should the most appropriate formulation have? Do any of the formulas analysed come close to this ideal?

Total carbohydrates content was determined as difference subtracting the protein, ash, moisture, and lipids contents from 1000? Is this correct?

How many training sessions were held before the qualification sessions? How was the panel performance validated? What type of scale was used for the evaluation? How many points did the scale have?

Are the anchors mentioned in the table real products, photographs or what exactly? Or are they just definitions in all cases? Please clarify

What is the difference between overall intensity of odor and meat-like odor? Only related attributes in meat are included in the description of both parameters. Overall intensity is usually defined as intensity of odor regardless of the type of odor. The same comment applies to taste overall intensity

Sweetness should be defined in a similar manner as the other odor notes “Association with…”

The authors should justify how it is possible to include in the same odour parameter chicken and pig.

The result section is only descriptive. It notes that as the raw material is different its characteristics are also different. The discussion about the suitability of each of the products tested for further application is very scarce.

Finally, the conclusions section is just a summary of the results. This section should be improved.

Round 2

Reviewer 4 Report

I recommend to accept the manuscript in the present form.